# Methodological Pitfalls of Investigating Lipid Rafts in the Brain: What Are We Still Missing?

**DOI:** 10.3390/biom14020156

**Published:** 2024-01-28

**Authors:** Kristina Mlinac-Jerkovic, Svjetlana Kalanj-Bognar, Marija Heffer, Senka Blažetić

**Affiliations:** 1School of Medicine, University of Zagreb, 10000 Zagreb, Croatia; kristina.mlinac.jerkovic@mef.hr (K.M.-J.); svjetlana.kalanj.bognar@mef.hr (S.K.-B.); 2Faculty of Medicine, Josip Juraj Strossmayer University of Osijek, 31000 Osijek, Croatia; 3Department of Biology, Josip Juraj Strossmayer University of Osijek, 31000 Osijek, Croatia

**Keywords:** cholesterol, detergent-resistant membranes, gangliosides, glycosphingolipids, Triton X-100, Brij O20, imaging mass spectrometry, neuronal membranes

## Abstract

The purpose of this review is to succinctly examine the methodologies used in lipid raft research in the brain and to highlight the drawbacks of some investigative approaches. Lipid rafts are biochemically and biophysically different from the bulk membrane. A specific lipid environment within membrane domains provides a harbor for distinct raftophilic proteins, all of which in concert create a specialized platform orchestrating various cellular processes. Studying lipid rafts has proved to be arduous due to their elusive nature, mobility, and constant dynamic reorganization to meet the cellular needs. Studying neuronal lipid rafts is particularly cumbersome due to the immensely complex regional molecular architecture of the central nervous system. Biochemical fractionation, performed with or without detergents, is still the most widely used method to isolate lipid rafts. However, the differences in solubilization when various detergents are used has exposed a dire need to find more reliable methods to study particular rafts. Biochemical methods need to be complemented with other approaches such as live-cell microscopy, imaging mass spectrometry, and the development of specific non-invasive fluorescent probes to obtain a more complete image of raft dynamics and to study the spatio-temporal expression of rafts in live cells.

## 1. Introduction

The organization of the plasma membrane has been intriguing researchers for decades. From a paper published in 1935 by Danielli and Davson [1] stating that “*there is now a considerable body of evidence supporting the view that living cells are surrounded by a thin film of lipoidal material*”, several paradigms were employed to describe the cell membrane. Through the Singer–Nicolson fluid mosaic model published in the 1970s [2], followed by the realization there are more specific, clustered membrane domains in the 1980s [3,4] and the development of the lipid raft (LR) concept in the 1990s [5], the outlook on the cellular membrane architecture evolved considerably. The everchanging perspective on membrane configuration and composition is inherently intertwined with the development of more elaborate, reliable methods that enable higher resolution and more insights into the function and behavior of membranes. Since cellular membranes are highly dynamic structures controlling the ultimate response of the cell to various signals and their surroundings, the methods we employ to investigate membranes and membrane domains need to be able to account for this dynamism, while at the same time acknowledging the specificities of individual membrane types. When it comes to the brain and neuronal membranes, it is also important to account for specific cell types and even membrane areas, e.g., to distinguish whether the post-synaptic or pre-synaptic membrane is analyzed. It is a difficult and challenging task of the researcher to choose the most suitable methods, critically evaluate the results, and reach the appropriate conclusions. This review is aimed to succinctly present an overview of the methodological approaches used in lipid raft research in the brain and to highlight potential pitfalls during the journey of navigating lipid rafts.

## 2. Structure and Function of Lipid Rafts

Cell membranes, structures thought to be composed of uniform lipid bilayers including randomly floated specialized components as proteins, carbohydrates, phospholipids and cholesterol, were accepted as the homogeneous fluid mosaic model of biological membrane proposed by Singer and Nicolson in 1972 [2]. Small, heterogenous, dynamic, and a specialized part of the external leaflet of the plasma membrane enriched with cholesterol and sphingolipids known as lipid raft microdomains (10–200 nm) have emerged as key players in orchestrating various cellular processes [6]. Occasionally, smaller rafts can be stabilized, resulting in the formation of larger platforms through interactions between raft proteins and lipids [7] or through interactions with surrounding binding partners, especially the components of the cytoskeleton [8]. The unique lipid-raft configuration establishes a microenvironment that is more organized and densely packed than the surrounding membrane. Lipid rafts are mobile and even though rafts exhibit a unique composition of proteins and lipids, there seems to be variability among rafts, with differences observed in both the proteins and lipids they encompass [9]. The size and composition of rafts are contingent upon the specific cellular membrane environment [10]. Sphingolipids, notably sphingomyelin, create a stable platform, and cholesterol functions as a molecular adhesive, enhancing the structural integrity of lipid rafts. Diverse cellular roles of lipid rafts are based on their specific molecular composition and functional features of each of the raft components (Table 1).

Keeping in mind that LRs are also involved in the formation of different vesicles, e.g., transport vesicles, synaptic vesicles, endocytic and exocytic vesicles [11], it came as no surprise that they are also present in intracellular membranes [12]. Even before the term lipid rafts was coined, in the 1990s they were described as glycosphingolipid-enriched, detergent-insoluble complexes [13] involved in the sorting and transport of cholesterol from the Golgi to the plasma membrane [14]. Apart from Golgi, LRs were described in other organelles as well, namely the endoplasmic reticulum (ER), mitochondria, mitochondria-associated ER membranes (MAMs), etc. [15,16,17]. MAMs in their entirety can actually be considered as lipid raft-like domains or intracellular lipid rafts that biochemically and physically connect mitochondria and the ER [18]. As such, they are hubs for ion transport and are implicated in regulating autophagy, and are being vigorously investigated in relation to various disorders [19,20].

**Table 1 biomolecules-14-00156-t001:** Molecular composition of lipid rafts and functional characteristics of raft components.

Raft Component	Functional Characteristic of a Raft Component
Lipids	
**Phospholipids**	A form fluid phospholipid bilayer made of saturated fatty-acid side chains of the phospholipids. This allows close packing with the saturated acyl chains of sphingolipids. Phospholipids provide flexibility and fluidity, allowing lateral movement of proteins and other molecules within the raft [21] and interact with cholesterol [22].
Phosphatidic acids (PA)	It contributes to the overall structural organization of lipid rafts affecting the fluidity and packing of lipid molecules within the raft [23]. In addition to that, PA is involved in regulation of membrane curvature [24], protein–lipid interactions within lipid rafts [23], and signaling modulation [25].
Phosphatidylcholines (PC)	Present in small amounts in lipid rafts [9]. Each of these phospholipids (PC, PE, PS, PI) contributes differently to the overall structure and function of lipid rafts, due to variability of both polar groups and saturation and length of fatty acyl residues in the hydrophobic backbone [26,27,28,29,30].
Phosphatidylethanolamines (PE)	
Phosphatidylserines (PS)	
Phosphatidylinositols (PI)	
**Sphingolipids** **(mostly sphingomyelin)**	Mostly present in the outer leaflet of plasma membrane [31]. Sphingolipids are integral to the structure and function of lipid rafts, contributing to their stability [32], organization, and involvement in signaling pathways [31]. The variations in the hydrophobic part of the molecule crucially contribute to lipid raft organization.
**Cholesterol and oxysterols**	Cholesterol plays a fundamental and multifaceted role in the structure and function of lipid rafts affecting structural integrity [21], modulating fluidity, organization and segregation of lipids and proteins within lipid rafts, modulation of cellular signaling pathways, endocytosis, and intracellular trafficking while cholesterol rich lipid rafts are often targeted by pathogens for cellular entry [21,22,32,33,34,35]. Oxysterols, derivatives of cholesterol that can either promote or inhibit the formation of lipid rafts, play a crucial role in signaling processes [36,37].
**Glycosphingolipids**	Gangliosides are sialylated glycosphingolipids, highly diverse and abundant in the mammalian brain. Gangliosides play important roles in various cellular processes, and their presence in lipid rafts has implications for the structure and function of these membrane microdomains [38,39,40].
**Proteins**	
**Raftophilic proteins**	
GPI-anchored proteins	Proteins anchored to the cell membrane via a GPI moiety, such as CD55, CD59 and Thy 1 are raftophilic and are involved in cell adhesion, signal transduction, immune regulation immune regulation, or neuronal connectivity [41]. The prion protein (PrP^C^) is also an important GPI-anchored protein, highly expressed in neurons that contain a sialic acid in their GPI-anchor structures [41]. Recent studies indicate PrP^C^as a key factor in cell fate regulation [42] and neuronal differentiation [43].
Receptor Tyrosine Kinases (RTKs)	Several RTKs, including the epidermal growth factor receptor (EGFR) and insulin receptor, preferentially associate with lipid rafts, influencing downstream signaling pathways [44].
G-Protein-Coupled Receptors (GPCRs)	Certain GPCRs, such as the serotonin receptor, are raftophilic. Their localization in lipid rafts impacts receptor signaling and cellular responses [45].
Src-Family Kinases	Src-family kinases, including Src and Lyn, are raftophilic proteins involved in signaling cascades and cellular processes such as proliferation and migration [46].
**Cholesterol-Binding Proteins**	Cholesterol-binding proteins contribute to the stability and functionality of lipid rafts [10].
**Intracellular Trafficking Proteins**	Proteins involved in vesicle trafficking and membrane transport are associated with lipid rafts. Their presence contributes to the regulation of endocytosis, exocytosis, and intracellular membrane dynamics [47].
**Immunoreceptors**	Immunoreceptors, including those on T and B cells, are known to be localized in lipid rafts. This localization is critical for efficient immune cell activation and response [48].
**Autophagy-Related Proteins**	The association of autophagy-related proteins (LC3, Beclin-1, ATG9, ULK1, WIPI) with lipid rafts depends on the cellular context, experimental conditions, and the specific phase of autophagy [49,50]. Additionally, the precise mechanisms through which lipid rafts influence autophagy and the functional significance of these associations are still areas of ongoing research.
**Apoptosis-Related Proteins**	Apoptosis-related proteins like Fas/CD95, caspases, Bcl-2 family proteins, DAPK, TNFR1, PrP, and FLIP were identified as associating with lipid rafts or being influenced by the organization of these membrane microdomains [42,51,52].

## 3. Lipid Rafts in the Brain

The investigation of the composition, organization, and functions of lipid rafts derived from mammalian brain is particularly demanding due to the immensely complex regional, cellular, and molecular architecture characteristic for the central nervous system in comparison with other tissues. One of the obstacles when dealing with the isolation of lipid rafts from brain tissue, which may lead to ambiguous conclusions, arises from the fact that lipid rafts reside in membranes of all cellular types in the central nervous system. The obvious limitations of experimental approaches utilizing classical biochemical methods are related to the heterogeneity of cell populations in brain tissue as well as to a large variety in the molecular composition, submembrane localization and dynamics of lipid rafts in live cells. Moreover, the morphology of brain cells shows high polarity which is, at the molecular level, related to concerted lateral submembrane structural and functional organization. This fine-tuned structural assembly, particularly of the neuronal membranes’ microdomains, seems to be crucial for the diverse cellular processes associated with neurodevelopment, maturation, and aging of the brain [53]. Despite plentiful methodological challenges, the significance and need for studying lipid rafts in the brain by multi-level innovative experimental approaches is strongly corroborated by the reported evidence as follows: (a) membrane surfaces of different brain cell types are huge and more elaborated than in non-neural tissues; (b) membrane lipids, especially the very ones constituting lipid rafts (e.g., gangliosides), are highly diverse and most abundant in mammalian brain; (c) lipid rafts are involved in a range of (patho)physiological processes occurring on cellular membranes [53,54]. Amongst various described functions, lipid rafts are assigned as cellular signaling platforms and well-regulated gathering points for specific proteins exerting actions vital for the brain [6,55]. Some of these actions include neurotransmission, synaptic plasticity, and membrane ion transport, which are practically all controlled and modulated by a local lipid environment through numerous dynamic intermolecular interactions [56]. The nature and precise occurrence of these interactions within lipid rafts in the brain has not yet been fully characterized. Everything mentioned speaks in favor of putting more effort into the research of spatio-temporal expression, distribution, and functions of brain lipid rafts, with the final goal of expanding our understanding of nervous system homeostasis at the molecular level.

## 4. Methodology behind Lipid Rafts: Biochemical Methods

The basis of the biochemical analyses of lipid rafts for the most part lay in their relative insolubility in cold non-ionic detergents. Since they have a distinctive lipid and protein composition, they are less fluid and segregate differently compared to the rest of the membrane. Therefore, following the extraction with non-ionic detergents, membranes separate into specific detergent-resistant membranes (DRMs), a term often used as a synonym for lipid rafts, and detergent-soluble membranes (DSMs), or the bulk (non-raft; nLR) membranes [57,58]. The most used detergents for LR isolation included Triton X-100, Brij O20 (previously called Brij 98), Brij 96, Lubrol WX, and CHAPSO. We could safely say that our raft knowledge, including the possible misconceptions, is largely defined by the biochemical methods of extracting and analyzing LRs. They set the tone and paved the way for all other investigations.

A typical protocol of an LR isolation [59] would include a homogenization step, the choice of which depends on whether cultured cells or tissue are used. That could be achieved by mechanically disrupting tissue in a homogenizer, sonication, passing the sample through a needle, etc. Usually, all the procedures are performed on ice or at +4 °C, in a buffer containing protease inhibitors and detergent or no detergent. This is followed by shorter centrifugation to remove cell debris, and ultracentrifugation, usually overnight, in discontinuous density gradients. Most often the gradients are obtained through overlaying sucrose solutions of different densities (*w*/*v*), but other ready-made reagents can be used. Finally, if there is a clear visible band of lipid rafts, individual fractions are collected and are ready for subsequent analyses. A generic overview of an assay using detergents is given in Figure 1.

The described isolation of LRs illustrated with these methods also entails the analysis of protein and lipid markers to evaluate their distribution. The consensus is that lipid-raft isolation can be considered as successful if the majority of the known accepted LR markers are indeed found in the LR fractions, and the most accepted bulk membrane markers are detected in the non-raft membrane fractions. A schematic representation of typical Western blot results performed on the fractions collected after LR isolation is given in Figure 2. In addition, protein concentration can be measured across all fractions with the highest concentration in lipid rafts. Furthermore, measuring cholesterol concentration is widely accepted as a method to pinpoint the rafts fractions since the concentration of cholesterol in those fractions is several times higher than the bulk membrane (Figure 2).

However, with the widespread use of detergents in LR analysis, apparent discrepancies in the results soon started to emerge [60,61]. Reports on disparate association of specific proteins and lipids to LR fractions exposed a methodological shortcoming—the differences in solubilization when various detergents are used, e.g., strong detergents like Triton X-100 vs. milder ones such as Brij O20. Even slightly different extraction conditions can produce significantly variable outputs. In order to pinpoint the real state of the raft organization that reflects physiological conditions accurately, individual biochemical methods need to be complemented by other approaches giving a complete and authentic picture of the membrane, e.g., single-fluorescent-molecule imaging in the live-cell [62], fluorescence resonance energy transfer (FRET) [63], imaging mass spectrometry (IMS) [64], etc. Nonetheless, a classical approach to LR extraction is still indispensable, depending on the aim of the researcher; e.g., if we want to structurally analyze the lipid composition of LRs in detail. With that in mind, the overview of the selected most widely used detergents as well as the detergent-free methods used for LR isolation are presented in Table 2. Detailed reviews of the selected methods and protocols, as well as comparison of the resistance of cell membrane to different detergents can be found in [59,61,65].

There are many disparities in the outcome of raft analysis based on the utilized detection methods. Whether or not detergent was used in the biochemical isolation is not the only point of possible divergence between results; it could be the affinity of the different antibodies used to detect raft or bulk membrane markers, the choice of the Western blotting method itself (e.g., wet, semi-dry, or dry transfer in Western blotting), the choice of markers based on the analyzed tissue or membrane, and whether additional antibodies, toxins, or fluorescent probes were used in the analysis. This is discussed in more detail in the following sections.

### 4.1. Cholesterol and Oxysterols

Undeniably, cholesterol plays a paramount role in lipid raft organization. In fact, in the common descriptions of lipid rafts, cholesterol and sphingolipids are highlighted as raft determinants [5,58]. Cholesterol concentration is highest in lipid rafts compared to the rest of the membrane (Figure 2), therefore, it is routinely determined in LR analysis. Even though colorimetry/spectrophotometry is a method of choice for determining serum cholesterol concentration, it is not sensitive enough for accurate measurement of cholesterol in raft fractions of neuronal membranes. For that purpose, the quickest and easiest approach to analyze cholesterol concentration is fluorometry [73], by using various commercial reagents utilizing enzyme-coupled reactions to produce highly fluorescent products which can readily be quantified. These assays can be adapted to measure both free cholesterol and cholesteryl esters.

A practically self-imposed question is how will the rafts and overall membrane dynamics be affected if we manipulate cholesterol levels? A very valuable tool in giving an answer to that question is cholesterol depletion, which has also become abundantly used as a means to study lipid rafts [35]. Reducing the concentration or completely removing cholesterol from the cellular membrane can be achieved by various methods, including removal of cholesterol from the membrane using methyl-beta-cyclodextrin, sequestration of cholesterol by cholesterol-binding compounds, or inhibition of cholesterol synthesis with statins [35,74]. Disrupting the optimal cholesterol levels has serious impact on the microposition and function of various proteins otherwise localized in LRs—cholesterol depletion assays were how raft-dependent pathways were identified [9]. Some of the receptors affected by cholesterol depletion include the EGF receptor, TrkA (NGF) receptor, PDGF receptor, and insulin receptor. Upon cholesterol depletion those receptors either change their microlocation, phosphorylation status, or both. In that way, many cellular signaling pathways can be affected [9].

Cholesterol metabolism is also affected by oxysterols, cholesterol oxidation products. Since their biophysical properties can be quite different from those of cholesterol, they also piqued interest in the context of LRs. However, oxysterols, of which there are many, are much more challenging to analyze than cholesterol itself [75]. The existence of many oxysterol species is not the only problem in their analysis. Their concentration is several orders of magnitude lower than cholesterol, so their isolation and characterization are severely overshadowed by cholesterol. Therefore, the methods used for oxysterol analysis include liquid chromatography (LC), gas chromatography–mass spectrometry (GC-MS), or LC-MS/MS methods [75,76]. The impact of oxysterols on the function of lipid rafts in brain development and aging is currently the most challenging question for future studies.

### 4.2. Raft Proteins and Bulk Membrane Protein Markers

As already mentioned, specific proteins are selectively localized in rafts vs. the bulk membrane [5,58], a fact quite useful for biochemically confirming LR isolation. In general, the proteins that cluster to LRs seem to do so through several mechanisms that include binding of cholesterol as an abundant LR component (such as caveolin) [77] and proteins with various lipid modifications [78,79]. GPI-anchored proteins are particularly abundant in LRs [9,80]. Some of the most commonly used proteins that we consider as raft markers are flotillin [66] and GPI-anchored proteins such as Thy-1 and PrP [9,59,81]. Therefore, their strong immunoreactivity in Western blotting following LR isolation is considered as proof that the LRs were isolated. At the same time, immunoreactivity of bulk membrane markers in non-LR fractions corroborates a technically “clean” LR isolation. Some of the bulk membrane markers include transferrin receptor (TfR) [59,66], amyloid precursor protein (APP) [64,71,72], and Na^+^/K^+^-ATPase (NKA) [82]. It must be stated that there is evidence that two pools of membrane NKA exist: the pumping pool outside of LRs where the majority of the present NKA resides, and the non-pumping NKA pool with signaling roles residing in rafts [83,84]. Therefore, one should also be mindful of the signaling events occurring in the cells at a particular moment, and if possible, always assess more than one raft and non-raft marker to ensure the optimal interpretation of the results.

### 4.3. Phospholipids

Lipid rafts, being enriched with glycosphingolipids, at the same time have lower content of (glycero)phospholipids compared to the rest of the membrane and whole cell preparations [9,58,85]. Apart from the sheer difference in lipid type preferentially present in lipid rafts, the composition of fatty acids present in lipids in rafts is different. LRs seem to harbor more saturated fatty acids which contribute to the more tight packing and organization of the rafts [58,86]. The tools used to investigate phospholipid composition in isolated lipid rafts using biochemical methods encompass purification of lipids by extraction with organic solvents, gel-filtration chromatography, ion-exchange chromatography, and usually separation with thin-layer chromatography (TLC) or high-performance thin-layer chromatography (HPTLC), followed by visualization. Visualization reagent depends on the lipid that needs to be detected. For phospholipids, molybdate reagent was often used [68,69,70,87]. In addition, phospholipids are readily analyzed by high-performance liquid chromatography (HPLC), ultra-performance liquid chromatography (UPLC), and mass spectrometry (MS) to obtain a detailed image of the composition of lipid rafts with all the structural details of the corresponding phospholipids. Furthermore, a metabolic labeling approach can also be employed, in which radioactively labeled metabolites (e.g., [^32^P]orthophosphate) are administered to live cells, and after incorporation of the radioactive compound, the LRs are isolated and analyzed for phospholipid composition and content by the methods already mentioned [87].

### 4.4. Sphingolipids

Sphingomyelin, classified as both phospholipid and sphingolipid, is far more enriched in LRs than other phospholipids [87] so it is considered in this section. Regardless of the fact that ceramides as well as sphingosine 1-phosphate have cellular roles on their own [88], in LRs, ceramides are primarily found as backbones of sphingomyelins and glycosphingolipids. Glycosphingolipids were introduced as the main LR determinants when the concept of lipid rafts emerged [5,89]. Amongst them, ganglioside GM1 is considered as a traditional raft marker and is often analyzed together with selected proteins in order to confirm biochemical LR isolation (Figure 2) [59,64,90]. Analysis of GM1, together with proteins by Western blotting, is possible due to the fact that it is recognized by cholera toxin subunit B (CTB) and can therefore readily be analyzed without the limitations of painstaking purification of gangliosides and detection via additional methods. Apart from using CTB to stain for GM1 in Western blotting, fluorescently-conjugated CTB is used to stain for lipid rafts in cells [91]. Furthermore, GM1 as the target ganglioside for LR detection serves as a probe precursor which can be modified into other fluorescent metabolites to detect LRs, such as BODIPY-GM1 [92].

However, if ganglioside analysis is a research goal, one has to be particulary mindful regarding the method of choice for isolating lipid rafts. Several studies have shown that using various detergents results in extremely different results in ganglioside composition. Studies complementing biochemical methods with immunohistochemistry and imaging mass spectrometry have proved Triton X-100 to be a severe disruptor of lipid rafts in the sense that it causes a misleading redistribution of membrane gangliosides and select GPI-anchored proteins. However, a milder Brij O20 seems to preserve the physiological distribution of gangliosides and should therefore be considered as a detergent of choice if ganglioside analysis is the aim. Since gangliosides generally pose an analytical challenge, having a hydrophobic ceramide backbone and a hydrophilic carbohydrate moiety, their analysis in lipid rafts is demanding. They tend to form mixed micelles with the detergents used in the isolation and it can be difficult to purify them. For that purpose, classical extraction with organic solvents is utilized, followed by DEAE anion-exchange chromatography and gel filtration [93]. After the purification gangliosides can be analyzed by HPTLC and visualized by resorcinol–HCl reagent [94]. However, even though gangliosides are extremely enriched in neuronal membranes, their concentration after LR isolation and purification is often too low for detailed detection using this method. Instead, immunoblotting (CTB overlay) can be used. In this method, gangliosides are resolved by HPTLC and overlaid with *V. cholerae* sialidase. The enzyme degrades complex gangliosides to GM1 and allows CTB detection of each ganglioside species with equal binding affinity for accurate direct quantitative comparison [95]. However, this analysis is limited only to complex gangliosides that can be degraded by *V. cholerae* to GM1 and therefore gangliosides that are not recognized by this enzyme and subsequently cannot be detected by CTB are omitted from the analysis. Of course, different mass spectrometry methods can be used to analyze raft gangliosides and structurally characterize them, but only if the extraction and purification of gangliosides from LRs is conducted in such a way that no residual contaminants, especially detergents which overshadow the MS spectra, remain in the sample.

Alternative methods for analyzing gangliosides in LRs include the utilization of specific monoclonal antibodies and analysis by different high resolution or super resolution microscopy methods. However, specific antibodies have been developed for the major most abundant ganglioside species, hence minor gangliosides will not be targeted with these analyses.

## 5. Visualizing Lipid Rafts

### 5.1. Antibodies, Toxins, and Fluorescent Probes

The main advantage of immunohistochemical/immunocytochemical methods is the ability to determine the tissue/cellular distribution of the molecule of interest. An overview of the most common mistakes arising from a misunderstanding of immunohistochemistry principles is presented in the work of Hoffmann et al. [96]. Antibodies conjugated with fluorophores are more suitable for multiplexed and high-resolution imaging compared to antibodies conjugated with enzymes [97]. It is important to note that highly specific antibodies are available for components of lipid rafts. For gangliosides, in addition to antibodies developed in chickens [98] and IgM class antibodies [99], mouse IgG antibodies are also available, which were developed in Galgt1 -/- knock-out mice deficient in the synthesis of all major gangliosides [100]. Antibodies targeting cholesterol have also been developed [101]. In cases where highly specific antibodies for lipid raft components are available, and fluorescent methods are employed, it is possible to quantify the colocalization (widefield, confocal, super-resolution, and microscopy) or even detect interaction of the molecules of interest (fluorescence resonance energy transfer—FRET). It is important to note that the majority of super-resolution microscopy techniques are better suited for studying cells and model membranes rather than complex tissues, such as the brain. Moreover, these methods are more commonly employed for the detection of fluorescent probes, rather than for the immunodetection of epitopes. Further improvement of existing microscopic methods depends on advanced image analysis software. In addition to algorithms designed for increasing resolution (deconvolution algorithms) and noise reduction of super-resolution images, software tools, that are intended for automatic segmentation of membrane structures, analysis of colocalization, or advanced software utilizing machine learning and deep learning techniques trained for the recognition and tracking of structures such as raftophilic molecules, will contribute to a further understanding of the nature of lipid rafts [96]. Furthermore, they will speed up image analysis and reduce the potential for human bias. How insights derived from the use of super-resolution microscopy techniques have expanded our current understanding of lipid rafts can be found in several excellent reviews [102,103,104].

The primary limitation of immunohistochemical methods in LR research is the resolution of the microscope, which is related to the wavelength of light and ranges from 200 to 300 nanometers laterally and 500 to 700 nanometers axially for fluorescent microscopes [105,106]. The resolution is slightly improved for confocal microscopes, thanks to the elimination of out-of-focus light (150–250 nm laterally and 500–700 nm axially), but is still insufficient for direct observation of lipid rafts whose size is estimated at 10–200 nm, depending on the cell type and the method (direct/indirect) used [9,107,108]. Among the techniques utilizing fluorophores for the detection of targeted molecules, only super-resolution microscopy allows for the direct exploration of lipid rafts due to lowering the resolution up to 10 nm [109,110,111]. The actual achieved resolution and the ability to localize individual fluorophores depend on various factors, including the quality of the fluorophores and the specific experimental conditions [112,113]. Among all available microscopic methods, immunoelectron microscopy offers superior resolution for the direct observation of lipid raft compositions. Two antibody labeling techniques are commonly employed: immunogold and immunoperoxidase, with transmission electron microscopy (TEM) being the prevalent imaging method. In addition to antibodies, labeled toxins with an affinity for GM1, other glycolipids, GPI-anchored proteins, or cholesterol have been also used [114,115]. The achievable resolution is constrained by the wavelength of electrons and the size of gold particles, or the electron-dense substrate for peroxidase, and ranges from 0.1 to 0.5 nanometers. Additional factors, including specimen preparation and antibody-related considerations, may further impact the achieved resolution.

The second major limitation of immunohistochemical/immunocytochemical methods stems from tissue fixation. The fixation (most commonly with paraformaldehyde) chemically alters the epitope or cross-links it to its functional ligand/receptor/enzyme or a molecule from the direct surroundings [116] so that the antibody does not recognize chemically the masked epitope. The impact of various fixation methods on the distribution and quantity of gangliosides, major components of brain tissue lipid rafts, is well illustrated in the study by Schwartz and Futerman from 1997 [95]. In addition to the selection of the most suitable fixative and avoiding excessively long fixation, it is possible to make its chemical reversal by the so-called antigen retrieval. The retrieval protocols are inherently aggressive and can lead to artifacts as they utilize high/low pH and heating, both of which are destructive to the structure of lipid rafts [117], whose recommended biochemical isolation strictly involves handling tissue at +4 °C. The next experimental mistake arises from the belief that fixation halts the mobility of molecules in the membrane [118]. Contrary to this, studies employing immunoelectron microscopy methods demonstrated clustering of molecules, along with an increase and stabilization of lipid bilayers induced by antibodies [115]. Therefore, for studies of lipid rafts, it would be more appropriate to use monovalent antibodies, which have only one antigen-binding site per antibody molecule, such as Fab fragments. Depending on the design of the experiment, clustering of membrane molecules induced by antibodies is not considered an artifact but rather desirable. In this way, an attempt is made to mimic the conditions of raft formation influenced by a physiological ligand or, for example, a virus targeting a specific lipid domain of the membrane [81,119,120,121]. Regardless of the use of divalent or monovalent antibodies, the optimization of the protocol for a given experiment begins with optimizing the type and duration of fixation. It should be sufficiently short to avoid the need for retrieval yet long enough to restrict the mobility of molecules in the membrane.

The third major limitation relates to the use of detergents inherent for lipid raft studies. The common belief that the use of detergents on fixed tissue enables the penetration of antibodies into the tissue, and at the same time does not lead to the redistribution or excessive washing out of molecules has been challenged by studies showing that under the influence of Triton X-100 [121], as well as most other detergents [122], at least glycolipids and GPI-anchored proteins are redistributed/lost. Redistribution occurs not only within the membrane of the same cell but also between membranes of different brain cells or even tissue sections that are incubated in the presence of detergent in the same well (free-floating immunohistochemistry method). For this reason, it is recommended to perform immunohistochemistry on lipid raft components without detergents, which excludes the detection of all detergent-resistant microdomains in intracellular organelles and intracellular epitopes of plasma membrane lipid rafts, including the main biomarker of lipid rafts—flotillins. Therefore, in many co-localization studies, ganglioside GM1—a molecule of the outer leaflet of the plasma membrane, which may be identified either with monoclonal antibodies or cholera toxin B subunit (CTB)—has been used as marker for rafts instead of flotillins. It is worth emphasizing that CTB has a high affinity for GM1, but it also binds to other glycolipids and glycoproteins [123,124]. Therefore, additional verification of binding specificity for the chosen experimental model is necessary. It is important to note that the GM1 signal (detected either by antibodies or CTB), in studies that did or did not use detergents, is primarily found within myelinated fibers and unexpectedly absent in the cerebral cortex of wild type animals [121,125,126,127], although all biochemical studies including mass spectrometry imaging studies (MSI) indicate that it should be present in the cerebral cortex [64,128,129]. Until the mechanism blocking the GM1 signal in the brain cortex is identified, it will pose a barrier to colocalization studies.

The fourth major limitation lies in the dynamic nature of lipid rafts, which can only be observed in live cells. The assembly and disassembly of lipid rafts in a sub-milisecond timeframe are influenced by various factors, including the interactions between lipids and proteins, membrane fluidity, interactions with cytoskeleton, and cellular signaling events. The technological challenge in live-cell microscopy lies in achieving a balance between spatial and temporal resolution. Pattern illumination super-resolution microscopy techniques, such as stimulated emission depletion (STED) and structured illumination microscopy (SIM), generally exhibit a faster image acquisition rate compared to single molecule localization microscopy (SMLM), including photoactivated localization microscopy (PALM), and stochastic optical reconstruction microscopy (STORM). This speed difference arises from the necessity in SMLM to capture and precisely localize individual molecules. Live-cell imaging generates a large amount of data, and in this case, further progress is expected through the application of software tools designed to automate data acquisition, storage, management, and analysis. An excellent overview of the challenges associated with observing lipid rafts in living cells is provided in the review by Nieto-Gari et al. [104]. Fluorescence resonance energy transfer (FRET), combined with confocal or super-resolution microscopy, have been instrumental in visualizing and studying the dynamic behavior of lipid rafts in real-time in living cells, but not well adopted for tissue [130]. In unfixed samples, artifacts caused by antibody-induced clustering can be expected whenever the acceptor and donor fluorophores are conjugated to antibodies in a FRET study. Therefore, instead of antibodies, fluorescent probes and fluorescently labeled toxins are more commonly used in live-cell microscopy, both of which have their limitations [104,131,132,133]. Recently, Kotani and colleagues developed a method utilizing CTB and the enzyme-mediated activation of radical sources (EMARS) reaction for the analysis of lipid rafts in live hippocampal slices [134] which opens new vistas in the research of lipid rafts of the brain.

The development of super-resolution microscopy has spurred the advancement of fluorescent molecular probes, including those designed for membrane components, a list of which was reviewed by Klymchenko and Kreder in 2013 [135]. Fluorescent molecular probes can be categorized into three main types: probes for lipid membrane components, probes with selective partitioning characteristics in the liquid ordered (Lo) or liquid disordered (Ld) section of the membrane, and environment-sensitive probes. In brain tissue research, the most commonly utilized class of molecules is the first one, with cholera toxin B (CTB) standing out, due to its binding to ganglioside GM1 used as a marker for lipid rafts [136]. Apart from the previously mentioned lack of specificity [123], a drawback of CTB is its induction of lipid-raft formation [137] and internalization commonly used for trans-neuronal tracking [138]. Other toxins that bind to lipid-raft components [139], such as Shiga toxin (binds to globotriaosylceramide), aerolysin (binds to GPI-anchored proteins), *Staphylococcus aureus* alpha-toxin (pore-forming toxin), or perfringolysin O (binds to cholesterol-rich membrane regions), have not found widespread application in brain histology and lipid raft studies, primarily due to their lack of specificity and destructive effects on lipid rafts. Unlike the previously mentioned toxins, which are protein molecules, filipins (filipin I, II, III, and IV) are polyene macrolide antibiotics isolated from the bacterium *Streptomyces filipinensis*, and they exhibit an affinity for cholesterol. Despite their application as histological markers, they are criticized for their poor fluorescent properties, disruption of lipid-raft structure, toxicity, and limited ability to distinguish between Lo and Ld domains of the membrane [140]. Probes belonging to the other two classes are more commonly employed in the investigation of model membranes than in the study of lipid rafts in tissues [135]. The main reasons for this lie in their distinct behavior under the conditions of complex natural membranes and their internalization. An exception is the small number of two-photon fluorescence turn-on probes developed primarily for FRET microscopy and visualising lipid rafts on living cells and tissue [131,141]. In functional studies, there is certainly a place for probes designed to label lipids with a fluorescent moiety, preferably through a long linker to avoid interference from the polar head in membrane interactions. Lipid anchors such as cholesterol and phospholipids are considered, as well as ceramide, as demonstrated in a study on Drosophila that revealed the role of ceramide (used as BODIPY labelled) in the exocytosis of synaptic vesicles [142]. It’s worth emphasizing that some fluorophores are non-fixable and, therefore, are intended solely for the investigation of live cells and tissues.

### 5.2. Direct Visualization

Mass spectrometry imaging (MSI) combines comprehensive analysis of various molecules (proteins, lipids, metabolites, drugs) with their spatial distribution in tissues. This multimodal imaging method does not require the use of antibodies or other types of probes that can be a source of artifacts. Tissue fixation is also not necessary, although it can be applied [143,144]. On the other hand, non-fixed tissue requires rapid processing; otherwise, there is a risk of microbiological contamination and degradation. MSI is ideal for the analysis of complex tissues such as the brain and for non-hypothesis-driven studies that generate big data. With the use of molecular standards, it can be quantitative and highly sensitive. For example, recently, MSI quantification of cholesterol was performed on sagittal sections of the wild-type mouse brain and *Npc1* null mouse (a model for Niemann-Pick type C1 disease) using on-tissue derivatization [145]. In previous attempts at quantification, lipid extraction from tissues [146], or the quantification of enzymes in cholesterol metabolism [147,148] was utilized, resulting in either the loss of spatial distribution or indirect conclusions.

The spatial resolution of MSI is related to the method of ion generation on the sample surface, so it depends on the diameter of the scanning laser in laser ablation, ion beam in ion beam bombardment, or size of droplets in droplet extraction [144]. Of the two laser focusing technologies, far-field and near-field, the recent iteration of the near-field desorption post-ionization time-of-flight mass spectrometer (NDPI-TOFMS) achieves a resolution of 250 nm [149] compared to the 600 nm resolution achieved with far-field technology like t-MALDI-2 [150]. Nevertheless, most commercial instruments utilize far-field technology and matrix-assisted laser desorption/ionization (MALDI), where desorption and ionization are accelerated by covering the sample or mixing it with a matrix. In MALDI technology, the laser spot size typically ranges from 1 to 20 µm, which is suitable for single-cell analysis but lacks sufficient resolution for lipid rafts.

The closest in resolution to the requirements for analysing lipid rafts is ion-beam-based technology, known as secondary-ion mass spectrometry (SIMS) [151]. SIMS utilizes high-energy ions (such as Cs^+^ or O^−^) as a primary ion beam for sputtering and ionizing molecules in the sample (referred to as secondary ions). The main drawbacks of this method are low ion yield and significant fragmentation. Gas cluster ion beams–SIMS (GCIB-SIMS) and the use of H_2_O clusters reduce fragmentation without compromising sensitivity at the expense of resolution [152]. This technique has been successfully applied to multi-omics analysis of frozen-hydrated neurons and brain sections [153] as well as for the deconstruction of cell membranes [154].

MSI is widely used for visualizing lipids in brain tissue [155], and the advancement of technology along with associated software places us on the verge of visualizing lipid rafts. Neurons, due to their size, serve as optimal models for single-cell and subcellular analyses, while 3D brain organoids are excellent models for observing the development of pathology.

The main criticisms of MSI undoubtedly include the high cost of the technology and its maintenance, artifacts caused by sample preparation (vacuum, freeze-drying) or matrix application (MALDI), complex analysis of results, low sensitivity for molecules that are less abundant in the sample, and limited accuracy in quantification. Nevertheless, a breakthrough in understanding the molecular complexity of the brain surely can be expected from utilizing MSI technology.

## 6. Conclusions

Biochemical fractionation, performed with or without detergents, is still the most widely used method to isolate lipid rafts. The subsequent analysis of lipid- and protein-raft constituents has been at the peak of the interest of researchers for decades. So, what are we still missing?

It is obvious that biochemical methods need to be complemented with other approaches to obtain a more complete image of raft dynamics. Artificial membranes and molecular simulation studies provide an additional level of insight into this complex organizational system. We must also keep in mind that rafts come in different “flavors” depending on their role and the transient conditions in the cell. It is becoming clear that there are many raft subtypes present even in the same cell [156]. Therefore, the danger of misinterpreting the research data on rafts lies in drawing too generalized conclusions from individual experimental conditions, and not including all types of raft constituents in the analyses we perform. Data cannot be interpreted correctly without taking into consideration cholesterol, sphingolipids, and specific proteins as well as their potential interplay. Therefore, even more interdisciplinary approaches will have to be adopted in order to be fully able to sail the lipid rafts.

## Figures and Tables

**Figure 1 biomolecules-14-00156-f001:**
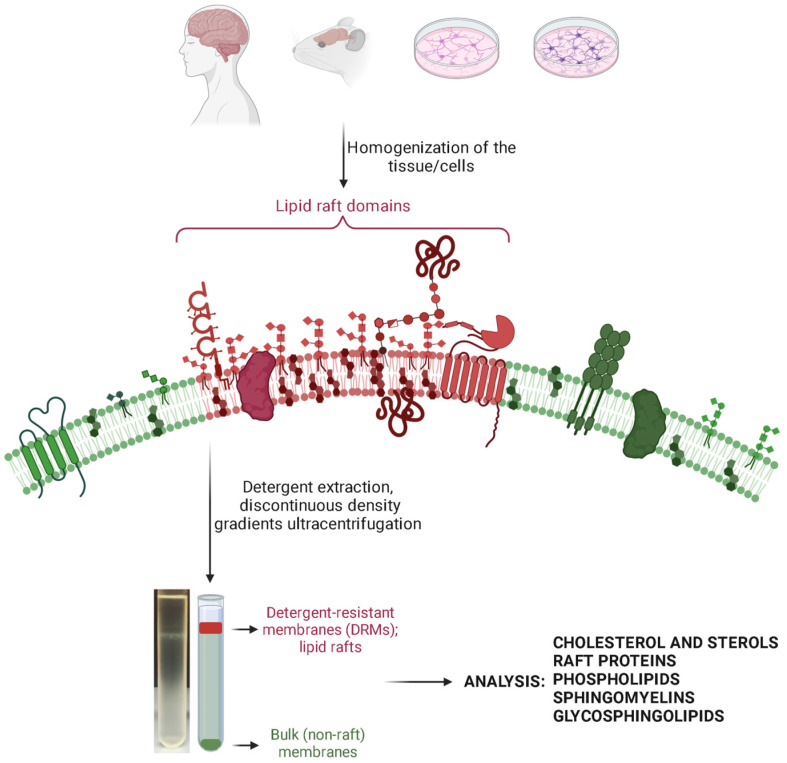
A generic overview of an assay using detergents for lipid raft extraction. The figure was created with Biorender.com (accessed on 14 December 2023).

**Figure 2 biomolecules-14-00156-f002:**
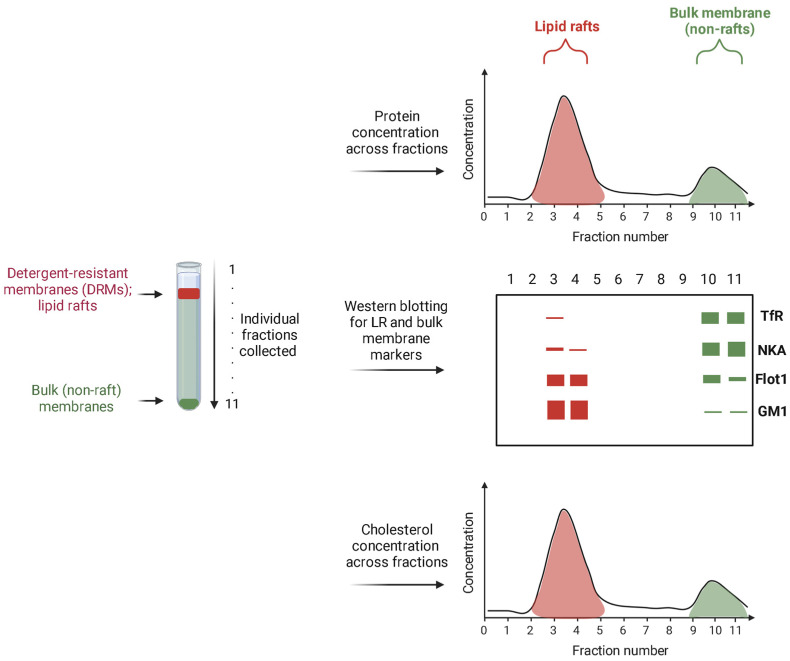
A schematic representation of the results of protein and cholesterol concentration measurement, as well as typical Western blot results performed on the fractions collected after LR isolation. The figure was created with Biorender.com (accessed on 14 December 2023).

**Table 2 biomolecules-14-00156-t002:** The summarized overview of selected methods of interest used for biochemical isolation of lipid rafts.

Material	Homogenization Method	Solubilization Method	Density Gradient	Analyzed Membrane Constituents			Reference
				Chol	Proteins	Lipids	
mouse brain	23-gauge needle	no detergent	sucrose and Optiprep^TM^	+	Flot1, TfR, MBP, GRASP65	−	[66]
mouse brain; HEK 293 cells	not specified; sonication	no detergent and Triton X-100	sucrose	+	caveolin-1, Flot1, TfR, GABA_A_ receptor subunits, NMDA receptor subunit NR1A	−	[67]
MDCK cells	Dounce homogenizer	Triton X-100	sucrose	+	PLAP, GPI-anchored proteins	PLs, SM, cardiolipin, LacCer, cerebrosides, sulfatides, gangliosides	[68] original method reporting the use of Triton X-100
mouse brain and cerebellar granule cells	Dounce homogenizer	Triton X-100	sucrose	+	Fyn, Lyn, PrP, Akt	PLs, SM, gangliosides	[69]
3A9 T cell hybridoma; mouse thymocytes; mouse T cells	sonication	Brij O20	sucrose	+	Thy-1, Lck, Rab-5	PLs, SM, gangliosides	[70] original method reporting the use of Brij O20
mouse brain	Potter–Elvehjem glass homogenizer with a Teflon pestle	Brij O20 and Triton X-100	sucrose	−	Flot1, TfR, GluA2, APP, Np65	gangliosides	[64]
CHO cells	23-gauge needle	Lubrol WX, Triton X-100 and CHAPSO	sucrose	+	Flot1, TfR, APP	−	[71]
primary rat hippocampal neurons	Dounce homogenizer	Lubrol WX and Triton X-100	Nycodenz^®^	+	CD71, clathrin, Flot1, PrP, APP	−	[72]

Chol: cholesterol; +: analysis was performed; −: analysis was not performed; Flot 1: flotillin1; TfR: transferrin receptor; MBP: myelin basic protein; GRASP65: Golgi reassembly and stacking protein 65; GABA: gamma-amino butyric acid; NMDA: *N*-methyl-D-aspartate; MDCK cells: Madin–Darby canine kidney cells; PLAP: human placental alkaline phosphatase; GPI: glycosylphosphatidyl inositol; PL: phospholipid; SM: sphingomyelin; Lac: lactose; Cer: ceramide; Fyn: a tyrosine protein kinase; Lyn: a Src family kinase; PrP: prion protein; Akt or PKB: protein kinase B; Thy-1 or CD90: Cluster of Differentiation 90, a GPI-anchored protein; Lck: a Src family kinase; Rab: Ras analog in brain; GluA2: glutamate receptor subunit 2; APP: amyloid precursor protein; Np65: neuroplastin 65; CHO: chinese hamster ovary.

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
