# Peer review of "Methodological Pitfalls of Investigating Lipid Rafts in the Brain: What Are We Still Missing?"

_biomolecules, 2024, doi:10.3390/biom14020156_

Round 1

Reviewer 1 Report

Comments and Suggestions for Authors

In this review entitled “Methodological pitfalls of investigating lipid rafts in the brain: what are we still missing?” the authors focus on the evaluation of biochemical and morphological methodologies used in lipid raft research in the brain.

Even if the present manuscript does not seem to add any novel insight, I recommend the publication of this work only after the authors have addressed the following major comment:

To improve the elucidation of lipid rafts, the authors should consider adding a concise explanation of microdomains in intracellular membranes. For instance, inclusion of details on mitochondrial rafts, rafts associated with the endoplasmic reticulum, and those associated with MAMs would be beneficial.  Additionally, it is crucial for the authors to emphasis on disparities in detection methods.

Furthermore the authors should add in the table the raft associated proteins involved in the autophagic and apoptotic processes.

Additionally, the authors should highlight the significance of Prp by indicating its enrichment not only in the lipid rafts of the plasma membrane but also within the Mitochondria-Associated Membrane (MAM) fraction after apoptosis triggering as suggested by several studies. doi: 10.1111/jnc.14891; doi:10.4161/pri.20479.

Comments on the Quality of English Language

Moderate editing of English language required

Author Response

Dear Reviewer 1, 

Thank you very much for your thoughtful and constructive feedback on our manuscript. We appreciate the time and effort you dedicated to reviewing our work. All changes made in manuscript are listed point-by-point in attached file.

We have highlighted all changes in the revised manuscript, and we believe these modifications have strengthened the overall quality of the paper. We remain committed to making any further revisions if necessary.

Thank you again for your valuable feedback.

Sincerely,

Senka Blažetić

Reviewer 2 Report

Comments and Suggestions for Authors

The article provides an overview of the methods used in lipid raft research in the brain and highlights the disadvantages of some research approaches. It highlights the challenges in studying neuronal lipid rafts due to their elusive nature and the complex molecular architecture of the central nervous system.

The practical implications of this work include the need for improved methods and interdisciplinary approaches to study lipid rafts in the brain, as well as the importance of considering specific lipid and protein components in their analysis. These implications can guide future research to understand the role and dynamics of lipid rafts in cellular processes and their potential influence on brain function and diseases.

Overall, this is a well-written review that provides a comprehensive overview of existing methods for lipid raft analysis in the brain.

The authors may shorten the sections under "Visualizing lipid rafts" (section 5).

Author Response

Dear Reviewer 2, 

Thank you very much for your thoughtful and constructive feedback on our manuscript. We appreciate the time and effort you dedicated to reviewing our work.

We have carefully considered your suggestion "The authors may shorten the sections under "Visualizing lipid rafts" (section 5)"

Minor corrections were made in the text to remove unnecessary repetitions. We apologize that the text was not further shortened - namely, the explanations requested by the third reviewer were added to it.

We have highlighted all changes in the revised manuscript, and we believe these modifications have strengthened the overall quality of the paper. We remain committed to making any further revisions deemed necessary.

Thank you again for your valuable feedback.

Sincerely,

Senka Blažetić

Reviewer 3 Report

Comments and Suggestions for Authors

This paper reviews methodologies employed in studying lipid rafts in the brain, emphasizing challenges and potential pitfalls. Lipid rafts, distinct membrane domains with specific protein and lipid compositions, play vital roles in cellular processes. Biochemical fractionation remains a common method, but discrepancies arise with various detergents. Complementary approaches, including live-cell microscopy and mass spectrometry, are essential for a comprehensive understanding. The study underscores the importance of considering membrane dynamics, cell types, and specific regions in neuronal membrane analysis.

  1. How can advancements in live-cell microscopy enhance the study of lipid rafts, considering their dynamic nature?
  2. What specific challenges arise in biochemical methods when studying lipid rafts in neuronal membranes compared to non-neural tissues?
  3. Could the paper elaborate on the potential impact of lipid raft organization changes induced by cholesterol depletion on cellular functions?
  4. Are there alternative methods suggested for ganglioside analysis in lipid rafts, given the limitations of biochemical approaches?

How can the use of advanced image analysis software contribute to improving our understanding of lipid raft interactions, especially in super-resolution microscopy?

Author Response

Dear Reviewer 3, 

Thank you very much for your thoughtful and constructive feedback on our manuscript. We appreciate the time and effort you dedicated to reviewing our work. All changes made in manuscript are listed point-by-point in attached file.

We have highlighted all changes in the revised manuscript, and we believe these modifications have strengthened the overall quality of the paper. We remain committed to making any further revisions if necessary.

Thank you again for your valuable feedback.

Sincerely,

Senka Blažetić

Round 2

Reviewer 1 Report

Comments and Suggestions for Authors

I accept in present form

Reviewer 3 Report

Comments and Suggestions for Authors

Authors have addressed my all questions very well. In have no further comments.